# Epidemiologic Trends of Thalassemia, 2006–2018: A Nationwide Population-Based Study

**DOI:** 10.3390/jcm11092289

**Published:** 2022-04-20

**Authors:** Jee-Soo Lee, Tae-Min Rhee, Kibum Jeon, Yonggeun Cho, Seung-Woo Lee, Kyung-Do Han, Moon-Woo Seong, Sung-Sup Park, Young Kyung Lee

**Affiliations:** 1Department of Laboratory Medicine, Seoul National University Hospital, Seoul 03080, Korea; jsleemd85@gmail.com (J.-S.L.); mwseong@snu.ac.kr (M.-W.S.); sparkle@snu.ac.kr (S.-S.P.); 2Department of Laboratory Medicine, Hallym University Sacred Heart Hospital, Anyang 14068, Korea; cygan83@naver.com; 3Division of Cardiology, Department of Internal Medicine, Seoul National University Hospital, Seoul National University College of Medicine, Seoul 03080, Korea; imcrtm@gmail.com; 4Department of Laboratory Medicine, Hangang Sacred Heart Hospital, Seoul 07247, Korea; pourmythe45@hallym.or.kr; 5Department of Statistics and Actuarial Science, Soongsil University, Seoul 06978, Korea; ghdk32@naver.com (S.-W.L.); hkd917@naver.com (K.-D.H.); 6Department of Laboratory Medicine, Hallym University College of Medicine, Chuncheon 24252, Korea

**Keywords:** thalassemia, prevalence, incidence, comorbidity, nationwide population-based study

## Abstract

Thalassemia is the most common form of hereditary anemia. Here, we aimed to investigate the 13-year trend of the epidemiologic profiles and risk of comorbidities in thalassemia using a nationwide population-based registry in Korea. Diagnosis of thalassemia, the comorbidities and transfusion events in patients with thalassemia were identified in the Korean National Health Insurance database, which includes the entire population. The prevalence of thalassemia increased from 0.74/100,000 in 2006 to 2.76/100,000 in 2018. Notably, the incidence rate has nearly doubled in the last 2 years from 0.22/100,000 in 2016 to 0.41/100,000 in 2018. The annual transfusion rate gradually decreased from 34.7% in 2006 to 20.6% in 2018. Transfusion events in patients with thalassemia were significantly associated with the risk of comorbidities (diabetes: odds ratio [OR] = 3.68, 95% confidence interval [CI] = 2.59–5.22; hypertension: OR = 3.06, 95% CI = 2.35–4.00; dyslipidemia: OR = 1.72, 95% CI = 1.22–2.43; atrial fibrillation: OR = 3.52, 95% CI = 1.69–7.32; myocardial infarction: OR = 3.02, 95% CI = 1.09–8.38; stroke: OR = 3.32, 95% CI = 2.05–5.36; congestive heart failure: OR = 2.83, 95% CI = 1.62–4.97; end-stage renal disease: OR = 3.25, 95% CI = 1.96–5.37). Early detection of comorbidities and timely intervention are required for the management of thalassemia.

## 1. Introduction

Thalassemia, the most common form of hereditary anemia, is caused by the impaired synthesis of one of the two globin chains in hemoglobin [1]. This disorder has been found to be highly prevalent in tropical and sub-tropical regions of the world (e.g., Southeast Asia, the Mediterranean area, the Indian subcontinent, and Africa), where the estimated prevalence rates are 12–50% in the case of alpha thalassemia and 1–20% in that of beta thalassemia [2,3].

Recently, the number of thalassemia patients has been increasing in other regions worldwide, including in North America, Northern Europe, and Northeast Asia, due to an increase in migrant populations, but there is a lack of comprehensive knowledge regarding the epidemiologic profile of thalassemia in these regions. A precise estimation of the prevalence and incidence rates of thalassemia is required for the effective management of patients, development of prenatal diagnostic strategies, and utilization of healthcare resources.

In South Korea, the National Health Insurance (NHI) program, a government-managed, single-payer, compulsory health insurance coverage for 100% of the Korean population (50 million in 2019), has been implemented [4,5,6]. Therefore, the NHI database provides comprehensive information regarding medical records, diagnoses, laboratory tests, prescriptions, and procedures. We hypothesized that patients with thalassemia in South Korea would be well covered by the NHI program; thus, the NHI database could adequately reflect the prevalence and incidence of thalassemia and risk of comorbidities in patients with this disorder.

In this study, we aimed to evaluate the 13-year trend of the nationwide prevalence and incidence of thalassemia in South Korea and to investigate the proportion of patients who received a single transfusion during the follow-up period and their risk of comorbidities.

## 2. Materials and Methods

### 2.1. Subjects and Database

This study was based on data recorded in a database of the Korean National Health Insurance Service (NHIS) from 2006 to 2018; the average population during this period was 51,104,424. The Korean NHIS offers medical coverage to 97% of the population, and the low-income group that makes up the remaining 3% is covered by Medical Aid. The medical claims database includes demographic information; monthly insurance premium (i.e., household income level); disease and comorbidity codes assigned to medical diagnoses based on the International Classification of Disease, Tenth Revision, Clinical Modification (ICD-10-CM); procedures; and prescription records. The study was approved by the Institutional Review Board of Hallym University Sacred Heart Hospital (IRB No. HALLYM 2019-12-011).

### 2.2. Definition of Thalassemia and Other Comorbidities

Patients diagnosed with thalassemia (D56 ICD-10-CM) were analyzed in this study. We identified patients with comorbidities based on whether the following disease codes were included in their records, regardless of the date of diagnosis: diabetes (E11, E12, E13, E14), hypertension (I10, I11, I12, I13, I15), dyslipidemia (E78), atrial fibrillation (I480, I481, I482, I483, I484, I489), myocardial infarction (I21, I22), stroke (I63, I64), heart failure (I50, I11.0, I50, I97.1), and end-stage renal disease (N185, Z49, N18, N19, Z905, Z94, Z992).

### 2.3. Transfusion

Transfusions of whole blood and red blood cells (RBCs) were identified based on the following codes: X1001 and X1002 for whole blood and X2021, X2022, X2031, X2032, X2091, X2092, X2131, X2132, X2512, X2111, X2112, and X2515 for RBCs. Patients who received one or more blood transfusions during the follow-up period were categorized as the transfusion group.

### 2.4. Income Variables

We estimated the financial status of each individual based on the monthly insurance premium that was paid. We classified the population into four groups, ranging from the first quartile (lowest income group) to the fourth quartile (highest income group), based on the amount paid toward the monthly insurance premium. Medical aid beneficiaries (3% of the total population) were categorized as the first quartile.

### 2.5. Statistical Analysis

Age-specific prevalence and incidence rates (stratified by 10-year age categories) of thalassemia were estimated in the sex-specific population. The risk of comorbidities in the transfusion group and transfusion-naïve groups was evaluated using logistic regression analysis. All statistical analyses were performed using SAS version 9.4 (SAS Institute, Cary, NC, USA). The *p* values presented are two-sided, and *p* values < 0.05 were considered to indicate statistical significance.

## 3. Results

The number of patients with thalassemia continuously increased from 354 in 2006 to 1577 in 2018 (Table 1). The prevalence rate of thalassemia also increased from 0.74 per 100,000 persons in 2006 to 2.76 per 100,000 persons in 2018. A subset of patients could be further categorized into α-thalassemia or β-thalassemia based on the ICD-10 codes (Appendix A). Overall, the prevalence of thalassemia in female patients was consistently higher than that observed in male patients (female-to-male ratio, 1.70–2.00) (Figure 1A). However, the trends in the incidence rate of thalassemia varied over time, decreasing from 0.24 per 100,000 persons in 2006 to 0.15 per 100,000 persons in 2011 but increasing from 0.15 per 100,000 persons in 2011 to 0.41 per 100,000 persons in 2018 (Table 2, Figure 1B). It is of note that the incidence rate has doubled in the last 2 years (from 0.22 per 100,000 persons in 2016 to 0.41 per 100,000 persons in 2018). During the study period, the incidence of thalassemia was higher in women than in men. Although the prevalence rate of thalassemia increased as the age of the population increased (Figure 2A), the incidence peaks were evidently high in the age ranges of 0–9 years, 30–39 years, and over 70 years (Figure 2B).

We further examined the difference in the prevalence of thalassemia according to the income level. In 2018, the prevalence was 2.93, 2.52, 2.91, and 3.64 per 100,000 persons in the case of the first (lowest income), second, third, and fourth quartiles (highest income), respectively (Figure 3).

The overall proportion of patients who received one or more blood transfusions was 27.7% (379/1366, 2006–2017) (Figure 4A). The annual transfusion rate gradually decreased: 34.7% (123/354) in 2006, 35.4% (161/455) in 2007, 33.5% (182/543) in 2008, 32.8% (206/628) in 2009, 31.9% (223/698) in 2010, 30.8% (235/764) in 2011, 31.4% (262/835) in 2012, 30.2% (283/936) in 2013, 29.0% (300/1036) in 2014, 27.7% (316/1139) in 2015, 25.6% (316/1234) in 2016, 23.6% (326/1383) in 2017, and 20.6% (325/1577) in 2018 (Figure 4B).

Interestingly, compared with in transfusion-naïve patients, the incidence of comorbidities was higher in patients who received blood transfusion (diabetes, 6.69% vs. 20.84%; hypertension, 16.51% vs. 37.73%; dyslipidemia, 10.03% vs. 16.09%; atrial fibrillation, 1.32% vs. 4.49%; myocardial infarction, 0.71% vs. 2.11%; stroke, 3.34% vs. 10.29%; congestive heart failure, 2.53% vs. 6.86%; end-stage renal disease, 3.04% vs. 9.23%; Appendix A). These comorbidities were significantly associated with transfusion events in thalassemia (diabetes: odds ratio [OR] = 3.68, 95% confidence interval [CI] = 2.59–5.22, *p* < 0.001; hypertension: OR = 3.06, 95% CI = 2.35–4.00, *p* < 0.001; dyslipidemia: OR = 1.72, 95% CI = 1.22–2.43, *p* = 0.002; atrial fibrillation: OR = 3.52, 95% CI = 1.69–7.32, *p* = 0.001; myocardial infarction: OR = 3.02, 95% CI = 1.09–8.38, *p* = 0.034; stroke: OR = 3.32, 95% CI = 2.05–5.36, *p* < 0.001; congestive heart failure: OR = 2.83, 95% CI = 1.62–4.97, *p* < 0.001; end-stage renal disease: OR = 3.25, 95% CI = 1.96–5.37, *p* < 0.001) (Figure 5).

## 4. Discussion

This is the first study to investigate the trends in the nationwide epidemiologic profiles of patients with thalassemia and their risk of comorbidities in the entire Korean population, spanning over 13 years. The prevalence of thalassemia has been increasing in the Korean population (from 0.74 per 100,000 in 2006 to 2.76 per 100,000 in 2018). Notably, the incidence rate has nearly doubled in the last 2 years from 0.22 per 100,000 in 2016 to 0.41 per 100,000 in 2018. This finding is comparable to that in a previous report from the United States and Northern Europe, where an increase in the prevalence of thalassemia was observed over the last 5 decades [3,7,8]. This increase in prevalence has been reported to be due to the influx of immigrants, which is probably one of the major causes of the increase in the thalassemia prevalence in South Korea. The number of foreign residents in South Korea has increased from 1.8% in 2006 to 4.6% in 2018. In 2018, 45.2%, 8.4%, and 8.3% of the foreign residents in South Korea immigrated from China, Thailand, and Vietnam, respectively [9]. In a recent prospective multicenter study conducted in South Korea, it was found that 8.3% of multiethnic individuals showed abnormal findings in hemoglobin electrophoresis and/or RBC indices, and 3.4% of them were diagnosed as being thalassemia carriers, as confirmed by genetic testing [10]. In addition, thalassemia has often been reported to affect Korean children from multicultural families or foreign-born immigrants in South Korea [11,12].

Three different peak ages (age groups: <10 years, 30–40 years, and >70 years) of patients with thalassemia have been identified, which may be because each subtype of thalassemia varies in severity and the onset of clinical manifestations. Individuals with β-thalassemia major/intermedia or hemoglobin H (HbH) disease usually exhibit clinical manifestations and transfusion dependency within 6 years after birth, whereas individuals with thalassemia minor are usually clinically asymptomatic or sometimes exhibit mild anemia [13,14]. People in their 30s are at an age when they frequently get married and have children; thus, thalassemia may be detected through screening processes, such as medical checkups before marriage and genetic testing of family members after the birth of the affected child (Appendix A). The peak in patients who are 70 years or older might be due to the late detection of thalassemia minor individuals who required blood for any other comorbidities (e.g., ESRD) [15]. Otherwise, a subset of patients with β-thalassemia or HbH disease may first require transfusion, even at 70 years or older. Higher prevalence of thalassemia in female patients than in male patients may be due to the ascertainment bias, as females are prone to seek medical attention due to any problem related to anemia (e.g., pregnancy or menorrhagia).

High rates of comorbidities, including cardiac complications, diabetes, and stroke, were observed in transfusion group, which was comparable to the findings in the literature [16,17,18,19,20]. Cardiac complications, such as arrhythmia and heart failure, in transfusion-dependent thalassemia are related to iron overload in the myocardium and are associated with a high risk of morbidity and mortality [21]. Insulin resistance due to iron disposition in pancreatic islet cells increases the risk of diabetes in patients with transfusion-dependent thalassemia [22]. The high rate of stroke in transfusion-dependent thalassemia is considered to be a reflection of a more hypercoagulable state with defective RBCs in patients who require transfusions. Careful monitoring and effective iron chelation in patients requiring transfusion have been shown to reduce the risk of comorbidities and contribute to improving the prognosis of this disorder.

This study has several limitations. Insurance claims data from the NHIS database were used in this study: Different medical institutes may have different diagnostic protocol, and it was often not possible to assess the specific subtypes of thalassemia (i.e., α- and β-thalassemia) based on the diagnostic code. In addition, we were not able to further investigate the prevalence and incidence by ethnicity and genetic variants, which were not included in the claims database. In future studies, matching such data (e.g., ethnicity and genetic variants) with data in the NHIS database would aid in elucidating the epidemiologic profiles based on thalassemia subgroups and types of genetic variants. Another limitation is that the limited range of comorbidities focusing on fairly common diseases rather than thalassemia-related complications (e.g., extra medullary hematopoiesis, pulmonary hypertension, osteoporosis, and gall stone) was analyzed in this study. Another important limitation is that the frequency of transfusion and the prescriptions for iron chelating drugs that provided us information on transfusion dependency could not be analyzed in this study, which did not enable us to sub-classify the transfusion group into the transfusion-dependent group and the non-transfusion-dependent group. The clinical spectrum of thalassemia ranges from mild to severe clinical symptoms, and transfusion needs in patients with thalassemia minor may be for other comorbidities.

In summary, the nationwide prevalence and incidence of thalassemia have been increasing in South Korea. Although the annual transfusion rate in thalassemia has been gradually decreasing, patients who received transfusion carry a higher risk of comorbidities. Early detection of comorbidities and timely intervention in these patients are required for the management of thalassemia.

## Figures and Tables

**Figure 1 jcm-11-02289-f001:**
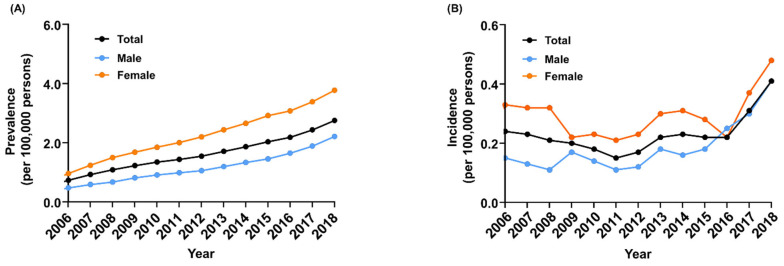
The (**A**) prevalence and (**B**) incidence of thalassemia in Korea, 2006–2018.

**Figure 2 jcm-11-02289-f002:**
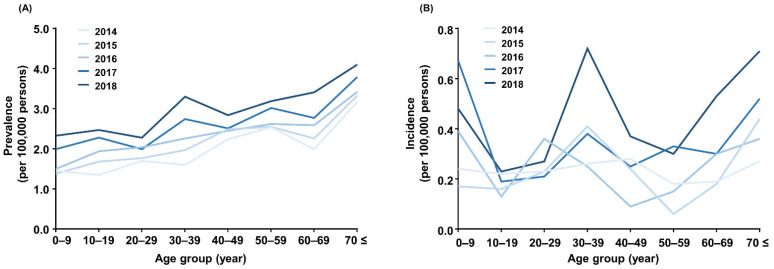
The (**A**) prevalence and (**B**) incidence of thalassemia in each age group.

**Figure 3 jcm-11-02289-f003:**
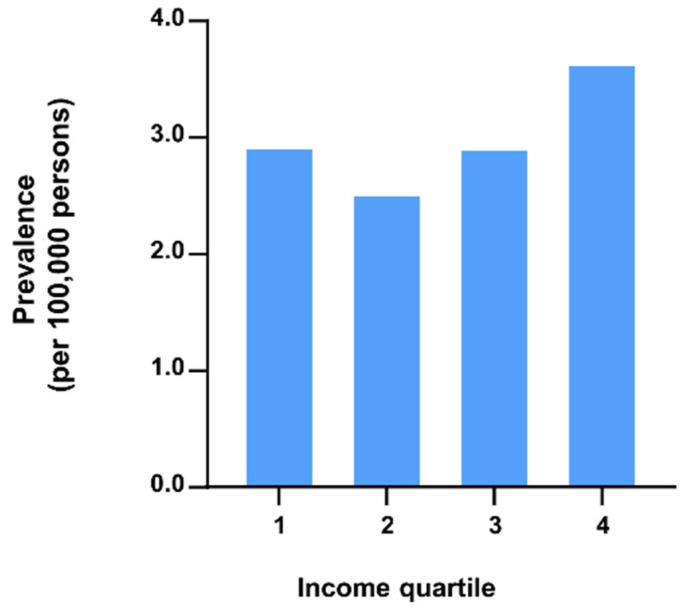
Thalassemia prevalence and incidence by income quartile, where quartile 1 is the lowest of the income distribution and quartile 4 is the highest.

**Figure 4 jcm-11-02289-f004:**
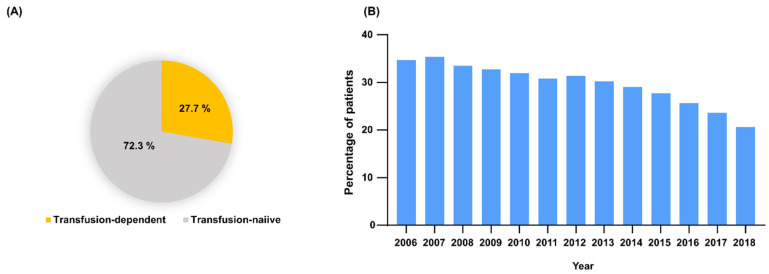
Blood transfusion in thalassemia. (**A**) Overall proportion of patients who received transfusions from 2006 to 2017. (**B**) Trend since 2006 of the percentage of patients who received a blood transfusion during study period.

**Figure 5 jcm-11-02289-f005:**
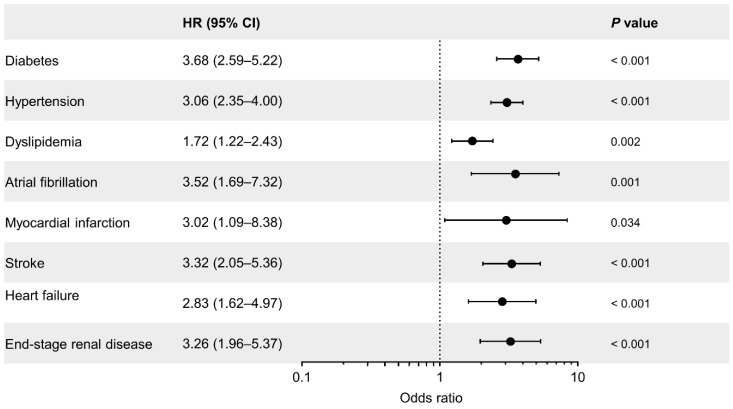
Odds ratio of a transfusion-dependent status in thalassemia for comorbidities.

**Table 1 jcm-11-02289-t001:** Actual numbers and prevalence of thalassemia in the Korean population (2006–2018).

Year	2006	2007	2008	2009	2010	2011	2012	2013	2014	2015	2016	2017	2018
Total population	49,238,227	49,672,388	50,001,057	50,290,771	50,581,191	50,908,645	51,169,141	51,448,491	51,757,146	52,034,424	52,272,755	52,426,625	52,556,653
Prevalence (/100,000 persons)													
Patients	354 (0.74)	455 (0.93)	543 (1.09)	628 (1.23)	698 (1.35)	764 (1.44)	835 (1.55)	936 (1.71)	1036 (1.87)	1139 (2.04)	1234 (2.19)	1383 (2.44)	1577 (2.76)
Sex													
Male	118 (0.48)	148 (0.59)	170 (0.68)	206 (0.82)	233 (0.92)	253 (0.99)	273 (1.06)	310 (1.2)	348 (1.34)	381 (1.46)	431 (1.65)	496 (1.89)	584 (2.22)
Female	236 (0.96)	307 (1.24)	373 (1.5)	422 (1.68)	465 (1.85)	511 (2.01)	562 (2.2)	626 (2.44)	688 (2.66)	758 (2.92)	803 (3.08)	887 (3.39)	993 (3.78)
Female-to-male ratio	2.00	2.10	2.21	2.05	2.01	2.03	2.08	2.03	1.99	2.00	1.87	1.79	1.70
Age group (years)													
0–9	41 (0.75)	49 (0.93)	50 (0.98)	59 (1.20)	65 (1.37)	68 (1.45)	65 (1.39)	62 (1.34)	67 (1.45)	64 (1.38)	69 (1.50)	89 (1.99)	101 (2.33)
10–19	25 (0.37)	30 (0.44)	37 (0.54)	40 (0.59)	51 (0.75)	58 (0.87)	67 (1.04)	81 (1.30)	81 (1.35)	96 (1.68)	106 (1.94)	121 (2.28)	127 (2.47)
20–29	33 (0.43)	47 (0.63)	53 (0.72)	49 (0.68)	52 (0.74)	64 (0.93)	81 (1.19)	97 (1.43)	116 (1.70)	122 (1.77)	142 (2.04)	139 (1.99)	160 (2.28)
30–39	57 (0.64)	67 (0.76)	81 (0.94)	101 (1.19)	105 (1.24)	107 (1.28)	102 (1.23)	108 (1.33)	128 (1.60)	155 (1.97)	175 (2.26)	208 (2.74)	248 (3.30)
40–49	71 (0.84)	93 (1.08)	120 (1.37)	136 (1.54)	152 (1.72)	153 (1.73)	166 (1.88)	182 (2.03)	201 (2.24)	222 (2.49)	218 (2.45)	221 (2.51)	244 (2.84)
50–59	55 (0.99)	75 (1.27)	94 (1.52)	104 (1.58)	116 (1.65)	144 (1.92)	159 (2.04)	192 (2.39)	210 (2.54)	214 (2.55)	223 (2.62)	259 (3.02)	279 (3.19)
60–69	35 (0.94)	48 (1.24)	49 (1.23)	63 (1.55)	72 (1.73)	75 (1.79)	77 (1.78)	83 (1.86)	94 (1.99)	115 (2.26)	140 (2.59)	158 (2.77)	205 (3.41)
70–79	37 (1.33)	46 (1.55)	59 (1.87)	76 (2.26)	85 (2.40)	95 (2.53)	118 (2.93)	131 (3.11)	139 (3.17)	151 (3.33)	161 (3.42)	188 (3.79)	213 (4.10)
Income													
Q1	79 (0.58)	119 (0.86)	150 (1.08)	176 (1.27)	220 (1.58)	244 (1.75)	231 (1.66)	267 (1.91)	280 (2.00)	296 (2.09)	342 (2.41)	361 (2.54)	418 (2.93)
Q2	83 (0.70)	88 (0.74)	92 (0.76)	116 (0.95)	111 (0.91)	127 (1.03)	161 (1.30)	203 (1.62)	208 (1.65)	231 (1.83)	249 (1.96)	274 (2.15)	322 (2.52)
Q3	89 (0.75)	123 (1.03)	149 (1.24)	156 (1.28)	164 (1.34)	175 (1.42)	190 (1.53)	188 (1.50)	227 (1.80)	267 (2.12)	279 (2.20)	323 (2.54)	372 (2.91)
Q4	103 (0.87)	125 (1.05)	152 (1.26)	180 (1.48)	203 (1.66)	218 (1.77)	253 (2.04)	278 (2.22)	321 (2.55)	345 (2.73)	364 (2.87)	425 (3.34)	465 (3.64)
Transfusion (%)	123 (34.7)	161 (35.4)	182 (33.5)	206 (32.8)	223 (31.9)	235 (30.8)	262 (31.4)	283 (30.2)	300 (29.0)	316 (27.7)	316 (25.6)	326 (23.6)	325 (20.6)

Q, quartile.

**Table 2 jcm-11-02289-t002:** Actual numbers and incidence rates of thalassemia in the Korean population (2006–2018).

Year	2006	2007	2008	2009	2010	2011	2012	2013	2014	2015	2016	2017	2018
Total population	49,238,227	49,672,388	50,001,057	50,290,771	50,581,191	50,908,645	51,169,141	51,448,491	51,757,146	52,034,424	52,272,755	52,426,625	52,556,653
Incidence (/100,000 persons)													
Patients	117 (0.24)	113 (0.23)	107 (0.21)	100 (0.20)	94 (0.18)	81 (0.15)	90 (0.17)	122 (0.22)	122 (0.23)	120 (0.22)	123 (0.22)	177 (0.31)	233 (0.41)
Sex													
Male	37 (0.15)	33 (0.13)	28 (0.11)	44 (0.17)	36 (0.14)	28 (0.11)	31 (0.12)	46 (0.18)	42 (0.16)	47 (0.18)	65 (0.25)	79 (0.30)	107 (0.41)
Female	80 (0.33)	80 (0.32)	79 (0.32)	56 (0.22)	58 (0.23)	53 (0.21)	59 (0.23)	76 (0.30)	80 (0.31)	73 (0.28)	58 (0.22)	98 (0.37)	126 (0.48)
Female-to-male ratio	2.20	2.46	2.91	1.29	1.64	1.91	1.92	1.67	1.94	1.56	0.88	1.23	1.17
Age group (years)													
0–9	15 (0.28)	10 (0.19)	5 (0.10)	11 (0.22)	12 (0.25)	9 (0.19)	11 (0.24)	8 (0.17)	11 (0.24)	8 (0.17)	18 (0.39)	30 (0.67)	21 (0.48)
10–19	6 (0.09)	7 (0.10)	5 (0.07)	3 (0.04)	9 (0.13)	8 (0.12)	0 (0.00)	7 (0.11)	13 (0.22)	9 (0.16)	7 (0.13)	10 (0.19)	12 (0.23)
20–29	9 (0.12)	13 (0.17)	13 (0.18)	7 (0.10)	9 (0.13)	9 (0.13)	14 (0.21)	19 (0.28)	16 (0.23)	16 (0.23)	25 (0.36)	15 (0.21)	19 (0.27)
30–39	20 (0.23)	19 (0.22)	16 (0.19)	14 (0.16)	14 (0.17)	10 (0.12)	11 (0.13)	18 (0.22)	21 (0.26)	32 (0.41)	19 (0.25)	29 (0.38)	54 (0.72)
40–49	25 (0.29)	22 (0.26)	29 (0.33)	20 (0.23)	21 (0.24)	15 (0.17)	19 (0.21)	27 (0.30)	25 (0.28)	21 (0.24)	8 (0.09)	22 (0.25)	32 (0.37)
50–59	18 (0.32)	19 (0.32)	14 (0.23)	8 (0.12)	9 (0.13)	11 (0.15)	14 (0.18)	15 (0.19)	15 (0.18)	5 (0.06)	13 (0.15)	28 (0.33)	26 (0.30)
60–69	11 (0.29)	12 (0.31)	9 (0.23)	16 (0.39)	7 (0.17)	9 (0.21)	6 (0.14)	8 (0.18)	9 (0.19)	9 (0.18)	16 (0.30)	17 (0.30)	32 (0.53)
70–79	13 (0.47)	11 (0.37)	16 (0.51)	21 (0.62)	13 (0.37)	10 (0.27)	15 (0.37)	20 (0.47)	12 (0.27)	20 (0.44)	17 (0.36)	26 (0.52)	37 (0.71)

## Data Availability

The data that support the findings of this study are available from the corresponding author, Y.K.L., upon reasonable request.

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
