# Peer review of "Epidemiologic Trends of Thalassemia, 2006–2018: A Nationwide Population-Based Study"

_jcm, 2022, doi:10.3390/jcm11092289_

Round 1

Reviewer 1 Report

Comments for the Author

This is a nationwide epidemiologic study in Korean patients with thalassemia. The study demonstrated the prevalence of thalassemia that has been increasing over time (from 2006 to 2018). The diagnosis of thalassemia was based on the ICD-10-CM (D56). The patients were classified into two groups including; 1) Transfusion-dependent thalassemia and 2) Transfusion-naïve thalassemia based on histories of red blood cell transfusion during the follow-up period. The study showed that the patients with transfusion-dependent thalassemia had higher rates of comorbidities (e.g. cardiac complications, diabetes, and stroke) than patients with transfusion-naïve. This study is interesting because it is a nationwide data over 13 years, however, some points can improve the manuscript as follows;

  • Comorbidities in this study are common diseases in the general population. It would be more interesting if the authors could report thalassemia-related complications, e.g., extra medullary hematopoiesis, pulmonary hypertension, osteoporosis, gall stone, etc.
  • There is a wide variety of clinical severity among patients with thalassemia. Patients with beta-thalassemia are more likely to have a severe phenotype than those with alpha-thalassemia. It would be more precise if the authors could categorize patients into beta-thalassemia or alpha-thalassemia based on the ICD-10 codes.
  • There are two groups of transfusion status in this study (transfusion-dependent and transfusion-naïve). However, most of the standard guidelines in thalassemia categorized patients as transfusion-dependent and non-transfusion-dependent thalassemia based on the frequency of RBC transfusion. It would be more accurate if the authors could divide the patients into three groups including; 1) transfusion-dependent, 2) non-transfusion-dependent, and 3) transfusion-naïve.

Reviewer 2 Report

The goal of this study as stated in the Introduction is as follows: “A precise estimation of the prevalence and incidence rates of thalassemia is required for the effective management of patients, development of prenatal diagnostic strategies, and utilization of healthcare resources.” This is a worthy goal as the authors later tell us that there is immigration to South Korea from China, Thailand and other countries with a known prevalence of thalassemia. They tell us that such immigrant populations have been demonstrated to have a certain percentage of thalassemia carrier individuals.

The methodology they used is via a national database from which they are able to find “comprehensive information regarding medical records, diagnoses, laboratory tests, prescriptions, and procedures.”

The authors found that the incidence doubled but the transfusion rate declined (number of patients rose from 354 to 1577, with more women than men.

They also found more transfusion events in individuals with diabetes, hypertension, A fib, MI, stroke and end stage renal disease.

Impressions:

The study is interesting but presents a somewhat inaccurate picture, it seems to this reviewer. This is because the authors included all individuals with a diagnosis of thalassemia (not specified if trait or with anemia) who have received at least one blood transfusion. No data was collected on whether or not the patient ever received a prescription for any of the known chelator drugs. Nonetheless they defined the receipt of even ONE transfusion as indicating “transfusion dependency” which is rather incorrect. No physician would classify any patient as being transfusion dependent based on having receive one transfusion. It is not clear to me why the authors did not simply use their extremely useful data base to look for prescriptions for iron chelating drugs, that would give them a much more accurate picture than a single transfusion.

The authors also identified comorbidities, some of which are indeed associated with TRUE transfusion dependent thalassemia (due to iron overload and the hypercoagulability of the chronic anemia with membrane abnormalities in thalassemia. So stroke, congestive heart failure, A fib are indeed more common in thalassemia patients due to iron overload however hypertension, MI and end stage renal disease are NOT comorbidities associated with thalassemia in its severe form. Therefore the authors have undoubtedly included some patients who have thalassemia trait but got a blood transfusion due to, for instance, end stage renal disease and not due to thalassemia and this skews the data. They identified more transfusion events in patients with diabetes, hypertension, A fib, MI, stroke and ESRD. At least some of these were due to the comorbidity and had nothing to do with thalassemia itself.

They also found more women than men with a diagnosis of thalassemia which may be due to the bias of ascertainment as women are more prone to seek medical attention due to any problem related to anemia for instance during pregnancy, or due to menorrhagia. Thalassemia has no sex predilection and the authors should say that this is ascertainment bias (unless they have another explanation).

Results: “the incidence peaks were evidently high in the age ranges of 0 – 9 years, 30 – 39 years, and over 70 years”. The Discussion says: Three different peak ages (age groups: < 10 years, 30 – 40 years, and > 70 years) of 45 patients with thalassemia have been identified, which may be because each subtype of thalassemia varies in severity and the onset of clinical manifestations. This is true but this is not the explanation for what they found. Beta thalassemia major (or HbE/thal) indeed generally presents in early childhood and Hb H disease or nontransfusion dependent beta thalassemia can present in middle age with a need for an occasional transfusion but the age peak of after 70 is probably due to spurious identification of thalassemia minor individuals who required blood for any of the comorbidities such as ESRD and were thus labelled “Thalassemia” patients. True some patients with nontransfusion dependent thalassemia such as HbH disease, or beta thal intermedia, may first require blood at an older age (and even may become transfusion dependent at that age). The possibility of detection at advanced age is certainly true even for thalassemia trait but the concept that individuals are detected at ages 30-40 due to the desire to get married and have children should be further justified by giving us the age at marriage in their country and the age at first childbirth. This sounds a bit late to this reviewer, I would think the peak would be a decade earlier for marriage and childbearing.

Discussion: The authors state: “Careful monitoring and effective iron chelation in patients with transfusion-dependent thalassemia may reduce the risk of comorbidities and contribute to improving the prognosis of this disorder”. This is incorrect. The word MAY is not true, it has been proven for decades that chelation improves life expectancy and the word MAY should be changed to “has been shown to…”. There are many references for this dating back decades.

Discussion: lines 70-71 the authors say that they could not discern the different subtypes such as beta or DELTA thalassemia!! They mean beta or alpha thalassemia. Delta thal is rare and only is important in erroneous diagnosis of individuals who are carriers for beta thalassemia (who therefore have low levels of HbA2), this state is rare and is not that significant in the context of a survey such as this. In contrast, alpha thalassemia is the type they needed to consider since this is the more prevalent type of thalassemia in the Far East.

In summary this reviewer suggests some modifications before acceptance, which should be feasible in view of the resources they have at their disposal.

Round 2

Reviewer 2 Report

The authors have revised and now if the reader gets to the Discussion, they do see the caveats that the requirement for transfusion may not be due to thalassemia. However the authors can go a bit further and clarify earlier in the manuscript that the need for transfusion was not necessarily due to the existence of a condition such as thalassemia minor. For instance, page 2 line 54 "patients REQUIRING TRANSFUSION" is misleading. The requirement for the transfusion may have had nothing to do with the thalassemia. While I can think of a way to say this properly, the authors may not like it. I would say something like: "patients who received a  single transfusion during the time period." If they have information on whether the individuals received multiple transfusions this would be important.

While this reviewer has published widely on the subject of thalassemia and recognizes its importance in the spectrum of global health concerns, it is not a good idea to publish distorted data. This could engender skepticism or even cynicism.
